# Return to work and sick leave patterns following a work injury among young adults: a study protocol of a Swedish multimodal study

Malin K Johansson ![ORCID] , Marie Hasselberg, Ritva Rissanen

## ABSTRACT

**Introduction** There is a lack of studies that focus exclusively on return to work (RTW) and sick leave patterns following a work injury among young adults. This study aims to close the gap by contributing with knowledge regarding young adults' sick leave pattern after a work injury and their experience of RTW after a work injury in Sweden.

**Methods and analysis** The present study is a multimodal study, which will use Swedish national register data and qualitative data collection by photovoice. Injuries classified as work injuries according to the Swedish injury classification were included. Registry data will be retrieved from the Swedish National-based registers of Swedish Information System on Occupational Accidents and Work-related diseases, the Swedish Social Insurance Agency's database MicroData for Analysis of Social Insurance and the Swedish Longitudinal Integration Database for Health Insurance and Labour Market Studies. Persons who have registered a work injury at the Swedish Work Environment Authority in 2012 will be included. Sick leave patterns will be analysed using group-based trajectory models and multivariate analyses to explore how sick leave patterns have developed over 5 years. Subsequently, a participatory approach using photovoice method will be conducted to explore young adults' perceptions of barriers and facilitators in RTW after a work injury. Results from the photovoice group sessions will be analysed using a grounded theoretical approach.

**Ethics and dissemination** The study has been reviewed and approved by the Ethical Review Board (case number 2019/028-6) in Sweden. Results of the study will be disseminated through peer-reviewed journals, print and media presentation, conferences and via reports to the funding agency.

Department of Global Public Health, Karolinska Institute, Stockholm, Sweden

**Correspondence to**
Dr Ritva Rissanen;
ritva.rissanen@ki.se

## Strengths and limitations of this study

► The use of registers offers a unique opportunity to capture sick leave patterns without a risk of bias as well as to provide access to assessments prior to the injury event.

► The use of a multimodal approach yields a more comprehensive knowledge base regarding the impact of a work injury among young working adults.

► Register-based analyses create an opportunity to map sick leave patterns and assess work injury with other age groups, as well as see how they vary depending on place of residence and socioeconomic status.

► Given the risks of using purposive sampling, one of the limitations we need to consider is the risk of research bias, which could lead to prior work injuries and other underlying factors being missed.

► The Swedish registers that are used in this study are mainly for administrative work and not for research; thus, we must consider that important variables relating to the research aim might be missing.

following a work injury. There are also some methodological weaknesses in these previous studies. From an international perspective, there is a paucity of registry studies within the field since the necessary registry data are typically not available. For instance, the majority of international research in the field have included participants who sought medical care for their injury,[3] or applied for financial compensation for their injury[4]; however, they are not using quality registers. Although high-quality registers can be found in, for example, Sweden, research using the Swedish registers have mainly focused on specific diseases such as the period after a heart attack, breast cancer and multiple sclerosis and not work injuries.[5–8] Thereof, studies using register data that shed light on young adults' sick leave pattern are scarce. To the best of the authors knowledge, no previous studies have been identified that focus on

## INTRODUCTION

In Sweden, the number of work injuries is increasing,[1] and the proportion of work injuries among young adults is increasing faster compared with other age groups.[2] Despite the high prevalence of work injuries among young adults, knowledge regarding young adults in working life is limited, as most international studies have investigated sick leave patterns targeting the general population

young adults' sick leave patterns using register data. To investigate the concept of sick leave pattern, we will focus on length of sick leave, including part-time sick leave independent of full return to work (RTW) during the follow-up period. Additionally, no qualitative studies exist on young adults' subjective perceptions in relation to a work injury, the period after the injury and RTW. Previous qualitative studies on work injuries have instead primarily focused on the general population's perceived promotion and inhibiting factors for RTW after specific type of injury,[9 10] as well as perceptions of employers or insurers on RTW.[11 12] Specifically, a previous qualitative study focusing on RTW among older people (50+) identified multiple RTW barriers at both individual and societal level such as being unfit to return to their previous employment area.[13] Furthermore, international studies on RTW have mainly dichotomised the outcome of return. That is, RTW has in most cases been classified either as RTW or not RTW, regardless of the degree or service the person has returned to.[3] This dichotomisation can also be observed in the few international studies that have used register data.[14] By categorising the outcome in such a way, research does not capture the nuances that RTW can include. Thus, RTW can happen gradually with shorter working days or recurrent sick leaves. By focusing on young adults in working life, a broader picture of RTW and the consequences of work injuries can be observed, which is important since young adults are generally beginners in the working life and can therefore be more vulnerable to the consequences of work injuries. RTW will be viewed as individuals rejoining the workforce regardless of employment contract.

It is important to increase knowledge on how young adults perceive their work injury and the consequences of the injury, as well as to identify what factors prevented or facilitated RTW in their own opinion. Specifically, it is important to study young adults in order to secure an enduring and sustainable workforce for the future. This paper presents the design of a multimodal study including a Swedish register-based cohort study and photovoice sessions on the impact of a work injury. The multimodal approach means that RTW is operationalised through sick leave pattern, whereas the first part of the study is a Swedish register-based cohort study on sick leave patterns that aim to:

► Assess the sick leave patterns for young adults following work injury in a Swedish setting.
► Assess how young adults' sick leave patterns differ from other age groups following a work injury.
► Investigate how sick leave patterns vary depending on sex, place of residence and socioeconomic factors.

The second part of the study consists of photovoice sessions on young adults' experiences of a work injury and aims to assess:

► How do young adults experience a work injury, the consequences of the injury and RTW?
► How do young adults experience factors that aim to prevent or facilitate RTW after a work injury?

## STUDY DESIGN AND METHOD

The study is a multimodal study which will use both Swedish register data and a qualitative approach with photovoice sessions. Planning for the study started in 2019, and the study is estimated to end in 2023. In 2012, approximately 30 000 work injuries were reported to the Swedish Work Environment Authority, of which about one-third were young adults (19–29 years). All injuries classified as work injuries according to the Swedish injury classification were included.[15] These include injuries due to unexpected events at work or road traffic injuries to or from their workplace but excluding work-related diseases. As the current study includes a register study, it is not possible to influence the size of the sample. However, the number of young adults in the Swedish registers who suffered a work injury with subsequent sick leave is considered to be large enough to determine any statistically significant differences in the analyses. For the second part of the study with photovoice sessions, a purposeful sampling will be utilised to assure an inclusion of participants with a variation in the sick leave patterns following a work injury. This type of purposeful sampling will ensure a study population with a diversity of knowledge of the consequences of the work-related injuries and preventive or facilitating factors of RTW following these injuries.

### Register study (study I)

We will use register data from the Swedish Information System on Occupational Accidents and Work-related diseases (ISA), the Swedish Social Insurance Agency's database MicroData for Analysis of Social Insurance (MiDAS), the Swedish Longitudinal Integration Database for Health Insurance and Labour Market Studies (LISA). The work injury will be identified in the ISA register, and data from ISA will consist of persons who reported a work injury in 2012. Work injuries that have been endured prior to the year of 2012 will not be taken into consideration. Each injury report had to be registered in the ISA register no later than 2 years postincident, since work injuries cannot be registered in the system more than 2 years in retrospect to the incident. Data from MiDAS and LISA will be collected for a 1-year period prior to the injury, that is, 2011 and for 5 years after the work injury (2012–2017) as it provides the possibility to control for sick leave before the work injury, and it allows for a long-term follow-up after the injury. By linking these Swedish registers, it will be possible to map out the impact of injuries on working life and how they differ depending on age, sex, place of residence and socioeconomic status.

### Photovoice study (study II)

Photovoice is a qualitative participatory method that requires participants to take photographs and use stories to identify issues of importance to the research aim. The participants will include young adults that have participated in the register study and have reported a work injury at the Swedish Work Environment Authority. The objective for using the photovoice methodology is that it

will include a varied selection where socio-demographic and work variables, type of injury, RTW and sick leave patterns are taken into consideration. A participatory approach will be used, drawing on a photovoice method to explore experiences of barriers and facilitators in RTW among young working adults. A participatory approach draws on the concept that persons are considered experts on their own life situation,[16] and in the present study, young working adults will be seen as important partners. In other words, the use of a participatory approach and photovoice methodology means that the photovoice members will be seen as experts in their own lives and thereof, will contribute with insight that others lack. The photovoice method mandates that a platform must be created where members can collectively reflect on their everyday life. In this process, documentation through pictures will be conducted; the pictures will thereafter be seen as a tool to enable change and empowerment.[17] The number of participants will depend on how many participants from the register study agree to partake. However, a minimum of five to eight participants should be included in each photovoice group.[18 19] We aim to include two to three photovoice groups in total.

## ELIGIBILITY CRITERIA (STUDY I)
### Inclusion criteria
The study includes all persons who have a registered work injury at the Swedish Work Environment Authority (ISA register) in 2012. In this study, a work injury is defined according to the Swedish regulation that states that a work injury is an injury that has occurred at the workplace or during travel to and from work. Hence, we include road traffic injuries according to this definition.

### Exclusion criteria
Work injuries that consist of a so-called occupational disease, that is, work strain that has occurred over time, as well as injuries that have resulted in death, will be excluded.

### Control group
Individuals that do not include the age group of young adults (age 19–29) will act as a comparison group to see if, and in what ways, young adults' sick leave patterns after a work injury differ from other age groups.

### Patient and public involvement
The public and patients were not involved in the design or planning of the study.

## MEASURES (STUDY I)
### Register data
Data from each registry will be retrieved by the register holders and merged by Statistics Sweden. Before delivery, the data will be deidentified where each individual receives a unique ID for the study. The key for reidentification is held for 3 months by the register holder and then destroyed. Thereafter, no reidentification is possible. Data will be collected on demographics, socioeconomic status (both persons and parents), work variables, injury variables and insurance variables from the Swedish Social Insurance Agency. By obtaining demographic and socioeconomic data, it becomes possible to identify vulnerable groups that might be hit harder by the consequences of a work injury. Furthermore, by analysing variables that are related to work, injuries and possible compensation from the Social Insurance Agency, it becomes possible to identify different sick leave patterns and variables (eg, type of injury) associated with these patterns.

### Data from ISA register
Data from ISA will consist of persons who have registered a work injury at the Swedish Work Environment Authority in 2012. Work injuries that were sustained prior to the year of 2012 will not be taken into consideration. However, we will include previous sick leave patterns in our analyses. The ISA register enables the collection of basic information and identification of individuals with a work injury. From ISA, information about participants (age, sex, occupation and type of employment), injury (injury mechanism and type of injury), employers and estimated absence will be collected. The included variables are age, sex, occupation, form of employment, type of injury, cause of work injury (deviation), contributing factor (external factor), injured body part, employer/nutrition branch and estimation on work absence (see table 1).

### Data from MiDAS register
Data from MiDAS will be collected from the years 2011–2017, which enables the analysis of a person's sick leave pattern. From MiDAS, data on sick leave will be collected, which provides an opportunity to analyse a person's social insurance pattern by following sick leave, the cause and days of sick leave, as well as sick leave spells following a work injury. The included variables are: sickness benefit, work injury compensation, rehabilitation benefit, variables related to parental allowance and unemployment benefits (see table 1).

### Data from LISA register
Data from LISA will be collected for the years 2011–2017, and the person who reported the work injury will be identified through the ISA register. From LISA, longitudinal data on the persons' work status, their health, working life, mobility and exit from work will be collected. The included variables are: id, date of death, municipal, civil status, country at entry/emigration, own country of birth, partners education level, household income, family status, highest education, finishing year for highest education, largest source of employment, workplace id, employment status, job number, change of employer, occupation, occupational based socioeconomic grouping, workplace municipality, branch office number, country of birth, father's country of birth, highest education for both

**Table 1** Information on register database, retrieved variables and year

| Type of register database and variables | Year | | | | | | |
|---|---|---|---|---|---|---|---|
| | 2011 | 2012 | 2013 | 2014 | 2015 | 2016 | 2017 |
| ISA | | | | | | | |
| Demographic characteristics | | X | | | | | |
| Occupational characteristics | | X | | | | | |
| Work injury characteristics | | X | | | | | |
| Date of injury | | X | | | | | |
| MiDAS | | | | | | | |
| Sick leave characteristics | X | X | X | X | X | X | X |
| LISA | | | | | | | |
| Socioeconomic status | X | X | X | X | X | X | X |
| Sickness benefit characteristics | X | X | X | X | X | X | X |
| Parents socioeconomic status | | X | | | | | |
| Immigration status | X | X | X | X | X | X | X |
| Date of death | X | X | X | X | X | X | X |

ISA, Information System on Occupational Accidents and Work-related diseases; LISA, Longitudinal Integration Database for Health Insurance and Labour Market Studies; MiDAS, MicroData for Analysis of Social Insurance.

parents, parents' profession and occupational socioeconomic grouping (see table 1).

## DATA ANALYSIS
### Register study (study I)
The analysis will include all individuals who reported a work injury to the Swedish Work Environment Authority during the calendar year of 2012, which is the exposure in the study. In addition, participants must have endured the work injury in 2012. Descriptive analysis will be conducted on age, age categories, sex, educational categories and types of injuries. The primary outcome of sick leave patterns will be analysed using group-based trajectory models (GBTM).[20] This type of analysis will be applied since it explores indications on how sick leave patterns have developed over time for each subgroup. We hypothesis that the results will indicate that several different trajectory groups will be assigned to GBTM, therefore, the one with lower Bayesian information criterion (BIC) and higher posterior class membership probability will be presented as the final pattern. The dependent variable is the mean number of sick leave days within a certain period (ie, 1–90 days) and for the two different research questions addressing sick leave patterns the independent variables are age, sex, place of residence and socioeconomic factors. Lastly, multivariate analyses such as binary logistic regression will be conducted to assess the association between sick leave patterns and age, sex, place of residence and socioeconomic factors. In the regression model, we will include injured body part and type of injury as predictors. When applicable, plausible confounders, that is, sex and socioeconomic status will be included in the analysis. OR and 95% CI will be used to describe the associations in the model. The analysis will only include cases with complete register data.

### Photovoice study (study II)
The analysis will be guided using a grounded theoretical approach, meaning a process of initial, focused and theoretical coding as well as comparisons between codes and data will be conducted.[21] A well-founded theoretical approach, such as the photovoice method, provides the opportunity to understand how young people's own experiences of the injury, the time after the injury and factors that prevent and facilitate RTW relate to each other and how they covariate.[21] To complement the grounded theoretical approach, a visual analysis[22] will also be performed in collaboration with the members of the photovoice group to identify possible barriers and possibilities in RTW after a work injury.

### Ethics and dissemination
Some of the ethical considerations that must be taken into account in the current project are informed consent and breach of privacy. The first part of the project is a registry study where information from ISA will be linked to information from MIDAS and LISA. This coordination will be done by the register authorities, and the research group will only have access to an unidentified database. All persons registered in ISA have also given consent for information about them to be stored in the register. The material cannot be linked to data sources other than those that were initially interconnected. The results will be presented at an aggregated level without possibility to identify individuals. For the second part of the project that includes photovoice sessions, informed consent will be obtained. However, there is a risk that participants will experience privacy infringement by making data collected

in the injury register available for a new study. To minimise this risk, participants will, in addition to the usual study information, be informed that the person cannot be identified in the register if they do not enrol in the photovoice sessions. In addition, the Swedish Work Environment Authority who holds the ISA register will send out invitations to ask if individuals with a work injury would like to participate in the study. If needed, counselling can be provided to the participants. The research group has previously conducted similar studies, which have demonstrated that people with injuries are often positive towards sharing information about their experiences in relation to the injury. There is no directly predictable benefit for the people participating in the study. However, in the long term, the results are expected to increase knowledge about the mechanisms behind RTW following a work injury for young adults. Dissemination of the results will be conducted through peer-reviewed journals, print media and internet, conference presentations and via reports to the funding agency. Ethical approval for the project has been sought and approved by the Regional Ethical Review Board in Stockholm (case number 2019/028–6).

## DISCUSSION

To the best of our knowledge, young adults' health outcomes after work injuries, their sick leave patterns and RTW are understudied areas since no studies have been identified that focus primarily on young adults.[23] This project, due to its use of the Swedish high-quality registries, creates a unique opportunity to study sick leave patterns and mechanisms behind RTW following a work injury among young adults. The photo stories will provide an effective means for the researcher to build an in-depth understanding of young adults' experiences of a work injury and RTW. The present project can, therefore, form a knowledge base for preventive measures and design of rehabilitation measures for young adults following a work injury. Furthermore, this knowledge base can minimise the time outside working life and provide opportunities for young adults to work with maintained good health throughout their careers. In other words, there is a significant social benefit from the study.

### Strengths and limitations

One of the main strengths of the study is the multimodal design by using a quantitative approach (registers) and a qualitative approach (participatory approach using photovoice method). The use of both methodologies will yield more data and a more comprehensive knowledge base regarding the impact of a work injury among young working adults that can be used to guide the design of preventative measures. This stepwise research process is considered important to yield a comprehensive description of work injuries. Identification of vulnerable groups can also be made, which is not possible internationally due to the lack of parallel registries. Instead, the Swedish register-based data create a possibility to map sick leave patterns and compare this with other age groups, including how they vary depending on place of residence and socioeconomic status. As a second step, a participatory

approach provides a person-centred perspective on work injuries and RTW without the influence from external stakeholders, such as service providers. By having a person-centred perspective where young working adults are considered experts, we hope to address a lack of representation in the working world and add a fresh perspective on work injuries that will hopefully lead to changes in RTW. Thus, the photovoice approach provides a platform for the target population, in which they will be able to raise awareness and possible solutions for problems that affect them.[17] Thereof, the first part of the study using Swedish registers might allow for a more unbiased assessment of sickness absence following work injury. Meanwhile, the second part of the study is intended for discussing personal solutions regarding RTW, but also for identifying and discussing specific challenges regarding the legal frameworks in the RTW process. This exemplifies how the stepwise contribution of each study part contributes to a comprehensive description.

There are also limitations with using Swedish National-based registers, whereby valuable information may be unavailable, inaccurate, unacquired or misclassified. Specifically, the registers are mainly used for administrative work and not for research; consequently, important variables relating to the research aim might be missing. For example, sick leave shorter than 14 days is not recorded in the register as the sick leave benefit is paid by the employer, which has implications on the analysis of short-term sick leave in close proximity to the injury event. Moreover, there might also be a lack of confounding information. In other words, difficulties may arise when trying to make causal inferences since information on potentially important confounding variables is only available at certain time points.[24] For instance, lacking information regarding socioeconomic factors, such as occupation, which is measured once a year, may lead to residual confounding.[25] However, the high-quality population-based Swedish registers are generally characterised as a relatively valid source of data. Swedish National-based registers generally include a broad range of variables, and register-based epidemiological studies are based on exact linkage by a unique personal number identification.[26 27] The specific number follows the specific individual throughout his or her life, and the same number is never given to a new individual,[28] which increases the validity of the registers. A limitation regarding the photovoice method is to recognise when the data collection should end due to saturation.[29] Yet, saturation is complex and almost impossible to achieve due to the diversity in human experiences.[29] Hence, the researchers will make the decision in relation to the number of participants and progress of data collection. The sample size in the first part of the project is predetermined due to the use of the Swedish registers. Meanwhile, the second part of the project is dependent on how many individuals from the Swedish registers are willing to take part in the photovoice sessions. The authors understand that there is a risk that the sample size might affect the reliability of the description of an event. Another limitation is that purposive sampling is prone to research bias. In this case, the researchers identify participants through register data; thus, prior work injuries and other underlying factors might

be missed. There is also the risk that participants might not understand how to use the photovoice method or that they might get tired of the sessions and withdraw their participation in the study. Accordingly, it is important that the authors make sure that the participants understand how to use the photovoice method, as well as making the participants feel engaged in the photovoice sessions.

The purpose of the study is to assess sick leave patterns among young adults and to determine how they might differ from other age groups, as well as to assess young adults' experiences of a work injury and RTW in a Swedish setting. The project also offers the unique opportunity to study outcomes on an individual level and is expected to provide knowledge that can form the basis for preventative measures and the design of rehabilitation measures for young adults following a work injury. In the long term, the information will provide a knowledge base on possible consequences of work injuries. Moreover, the information can be used as a basis for preventive measures and for the establishment of rehabilitation plans following a work injury to promote a healthy start to the career of young adults. Lastly, the results of the project will be disseminated through scientific publications, participation and presentations at conferences, both domestically and internationally.

**Contributors** RR conceptualised the project and MKJ drafted the manuscript. MH and RR reviewed and edited the manuscript. All authors read and approved the final manuscript.

**Funding** This work was supported by AFA insurance grant number 180 273. AFA insurance is not involved in the design, analysis or reporting of the funded studies.

**Competing interests** None declared.

**Patient and public involvement** Patients and/or the public were not involved in the design, or conduct, or reporting, or dissemination plans of this research.

**Patient consent for publication** Not required.

**Provenance and peer review** Not commissioned; externally peer reviewed.

**ORCID iD**
Malin K Johansson http://orcid.org/0000-0002-5165-1693

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
