## [Reviewer comments · BMJ Open]

ARTICLE DETAILS

TITLE (PROVISIONAL)	Return to work and sick leave patterns following a work injury among young adults: a study protocol of a Swedish multimodal study
AUTHORS	Johansson, Malin K; Hasselberg, Marie; Rissanen, Ritva

VERSION 1 – REVIEW

REVIEWER	Chuang, Hung-Yi Kaohsiung Medical University Hospital, Department of Occupational and Environmental Medicine
REVIEW RETURNED	19-Oct-2020

GENERAL COMMENTS	This study will research the young adults' sick leave pattern after a work injury and their experience of return to work (RTW) after a work injury using data from the Swedish registers. The protocol has both quantitative and qualitative methodology. The limitation might meet in the protocol was clearly presented. Two comments for the author may be helpful to clarify the study: They want to use the group-based trajectory model for sick leave patterns, then multivariate analyses. However, the more detail description of dependent and independent variables will be better for readers to understand their multivariate analyses. For the qualitative study part, the purposeful sampling should be clear to show what kind of injured workers will be selected or what trajectory pattern of sick leave will be selected.
---

REVIEWER	Burdorf, Alex Erasmus MC, Department of Public Health
REVIEW RETURNED	22-Oct-2020

GENERAL COMMENTS	1. The title correctly states the order of events, i.e. RTW and then sickness absence patterns, whereas in the abstract the changed order is confusing. it would also help to know the duration of follow-up after the RTW, and to distinguish between RTW and subsequent sickness absence patterns2. It would be helpful to present a definition on injury, is this is internationally a confusing term.3. Introduction: see also remark 1. It is helpful to make a clear distinctioio between RTW after the injury and subsequent sickness absence, as the latter reflects longterm consequences of the injury (and, a priori, one would expect large differences across types of injuries). It is not clear from the introduction nor from the study aims whether the study primarily addresses duration of sickness absence before full RTW or also sickness absence in the period
--

	thereafter. It appears only in the study design that longterm follow-up is also intended. 4. Design: why 2012 as year? I guess that has to do with the intended 5 years follow-up, but why is follow-up 5 years (i would expect that a 3 year follow-up is sufficient for the aims of the study) 5. Design: with so many databases involved, a table with database and retrieved variables would be helpful 6. the photovoice approach is interesting, but one would expect a minimum number of participants in this part of the study. 7. The control group seems to be relevant for duration of RTW, but for sickness thereafter i would advise to take a control group of young adults without an injury. For most injuries, I would hypothesize that there will no longlasting effects and, thus, sickness absence patterns will reflect these patterns among their peers 8. Type of injury seems an important factor, what is available? How is it diagnosed, and categorized? 9. Data analysis: see my previous remark. RTW process is entirely different than sickness absence later. I would expect two different analysis: (i) Cox analysis for RTW, with eg type of injury as factor, and (ii) trajectory models on sickness absence after RTW. 10. Discussion: there is no definition of sickness absence, but surely the register only has sickness absence after a certain number of days (10, 2 weeks?). This has implications.
--	---

REVIEWER	Flachs, Esben Bispebjerg Hospital, Occupational and Environmental Medicine
REVIEW RETURNED	27-Oct-2020

GENERAL COMMENTS	Thank you for letting me review the protocol. I have some comments for the authors to consider. 1) Sample size? It is mentioned twice in the protocol but never elaborated upon. What are the expectations of relevant differences in the register studies, and which numbers are required to investigate these with confidence. As mentioned it is not possible to expand the 2012 register, but it might be possible to include eg. 2013 in the investigations. On sample size in the qualitative research part it is written that: "Hence, the researchers will make the decision in relation to the number of participants and progress of data collection." But no criteria are listed for either numbers of participants or data collection. It would strengthen the protocol to make sure criteria for sample size are decided upon before start of the study. Statistical analysis: The authors state that in addition to "Group-based Trajectory Models" analysis will also include Multivariate analyses, but not which multivariate analyses (linear, logistic,...) and neither on which outcomes these analyses are applied.
--

	In the discussion on confounders: The authors state that: "When applicable plausible confounders will be included...". It would again strengthen the design if such decisions are taken in advance, thus which confounders are likely, and which are possible to include, and how will these be included. On the qualitative part of the study: I have no experience with qualitative methods, and is thus not able to review this part, except for the sample size considerations mentioned above. The ethics are thoroughly described. The written language could do with another thorough bypass. Limitations are somewhat generally/speculatively described, I would like a more focussed approach, I would think the authors are aware of which required or wanted information is missing in the registers.
--	---

REVIEWER	Cancelliere, Carol University of Ontario Institute of Technology
REVIEW RETURNED	02-Nov-2020

GENERAL COMMENTS	1. Select appropriate conduct and reporting guidelines for each study type (cohort, qualitative) and ensure all items are addressed. E.g., STROBE: Clearly define all variables and how they will be handled in the analysis, describe which choices of groupings were chosen and why, how will you address potential sources of bias, describe all statistical methods (what type of multivariable analysis), how will the outcomes be reported, will you report adjusted estimates with 95% CIs. etc. E.g. SRQR: Please address all items.
---

VERSION 1 – AUTHOR RESPONSE

Comments to the Author Reviewer 1 Dr. Hung-Yi Chuang, Kaohsiung Medical University Hospital, Kaohsiung Medical University	Answers from authors
They want to use the group-based trajectory model for sick leave patterns, then multivariate analyses. However, the more detail description of dependent and independent variables will be better for readers to understand their multivariate analyses.	The dependant and independent variables have been clarified in the manuscript, p. 9.
For the qualitative study part, the purposeful sampling should be clear to show what kind of injured workers will be selected or what trajectory pattern of sick leave will be selected.	A clarification has been added. Please see page 5.

--	--

Comments to the Author Reviewer 2 Prof. Alex Burdorf, Erasmus MC	Answers from authors
The title correctly states the order of events, i.e. RTW and then sickness absence patterns, whereas in the abstract the changed order is confusing. it would also help to know the duration of follow-up after the RTW, and to distinguish between RTW and subsequent sickness absence patterns	The order of the wording has been changed and the follow up time has been added to the abstract, p. 2.
It would be helpful to present a definition on injury, is this is internationally a confusing term.	A clarification of the definition of injury has been added to the abstract (p. 2.) and method section (p. 5).
Introduction: see also remark 1. It is helpful to make a clear distinctioio between RTW after the injury and subsequent sickness absence, as the latter reflects longterm consequences of the injury (and, a priori, one would expect large differences across types of injuries). It is not clear from the introduction nor from the study aims whether the study primarily addresses duration of sickness absence before full RTW or also sickness absence in the period thereafter. It appears only in the study design that longterm follow-up is also intended.	Thank you for this comment. This study will address sick leave patterns independent of full RTW. A clarification has been added to the introduction (p. 4).
Design: why 2012 as year? I guess that has to do with the intended 5 years follow-up, but why is follow-up 5 years (i would expect that a 3 year follow-up is sufficient for the aims of the study)	The year 2012 was chosen to allow for a five-year follow-up. The five-year follow-up period was chosen as previous studies indicate long-term sequel which expands beyond a three-year period.
Design: with so many databases involved, a tbale with databse and retrieved variables would be helpful	Thank you for the suggestion. A table has been added to the method section (p. 8-9).
the photovoice approach in interesting, but one would expect a mimimum number of participants in this part of the study.	A clarification of the minimum number of participants for the photovoice section has been added (p.7).

The control group seems to be relevant for duration of RTW, but for sickness thereafter i would advise to take a control group of young adults without an injury. For most injuries, I would hypothesize that there will no longlasting effects and, thus, sickness absence patterns will reflect these patterns among their peers	Thank you for this suggestion. However, due to limitations of this study it is not possible to include a control group of young adults without an injury rather a comparison will be made within the young adult group and the different reported trajectories including those who do not report any sick leave.
Type of injury seems an important factor, what is available? How is it diagnosed, and categorized?	A clarification of the definition of injury has been added to the abstract (p. 2.) and method section (p. 5).
Data analysis: see my previous remark. RTW process is entirely different than sickness absence later. I would expect two different analysis: (i) Cox analysis for RTW, with eg type of injury as factor, and (ii) trajectory models on sickness absence after RTW.	Thank you for the suggestion. We agree that a cox analysis with type of injury as a factor would be interesting to include. However, this study does not aim to investigate if different types of injuries influence on RTW and sick leave. Instead, we aim to assess young adults sick leave patterns in general after a work injury.
Discussion: there is no definition of sickness absence, but surely the register only has sickness absence after a certain number of days (10, 2 weeks?). This has implications.	A short section regarding this has been added to the discussion (p. 11).

Comments to the Author Reviewer 3 Dr. Esben Flachs, Bispebjerg Hospital	Answers from authors
Sample size? It is mentioned twice in the protocol but never elaborated upon. What are the expectations of relevant differences in the register studies, and which numbers are required to investigate these with confidence. As mentioned it is not possible to expand the 2012 register, but it might be possible to include eg. 2013 in the investigations. On sample size in the qualitative research part it is written that: "Hence, the researchers will make the decision in relation to the number of participants and progress of data collection." But no criteria are listed for either numbers of participants or data collection. It would strengthen the protocol to make sure criteria for sample size are decided upon before start of the study.	As mentioned in the protocol the sample size in the first part of the project is predetermined due to the use of nationwide Swedish registers. The sample size for the qualitative research part has been clarified in the protocol (p. 7).

Statistical analysis: The authors state that in addition to "Group-based Trajectory Models" analysis will also include Multivariate analyses, but not which multivariate analyses (linear, logistic,...) and neither on which outcomes these analyses are applied.	Thank you for the comment. The primary outcome and type of multivariate analyses has been included in the Analysis section (p. 9).
In the discussion on confounders: The authors state that: "When applicable plausible confounders will be included...". It would again strengthen the design if such decisions are taken in advance, thus which confounders are likely, and which are possible to include, and how will these be included.	Thank you for this comment. Possible confounders have been data analysis section page 9.
On the qualitative part of the study: I have no experience with qualitative methods, and is thus not able to review this part, except for the sample size considerations mentioned above.	Thank you.
The written language could do with another thorough bypass.	The protocol has been reviewed by a professional language reviewer.
Limitations are somewhat generally/speculatively described, I would like a more focussed approach, I would think the authors are aware of which required or wanted information is missing in the registers.	Changes have been made accordingly to the protocol (see p. 11).

Comments to the Author Reviewer 4 Dr. Carol Cancelliere, University of Ontario Institute of Technology, Canadian Memorial Chiropractic College	Answers from authors
Select appropriate conduct and reporting guidelines for each study type (cohort, qualitative) and ensure all items are addressed. E.g., STROBE: Clearly define all variables and how they will be handled in the analysis, describe which choices of groupings were chosen and why, how will you address potential sources of bias, describe all statistical methods (what type of multivariable analysis), how will the outcomes be reported, will	Please see comments to previous reviewers. We have followed STROBE guidelines and clarified accordingly.

you report adjusted estimates with 95% CIs. etc. E.g. SRQR: Please address all items.	
--	--

VERSION 2 – REVIEW

REVIEWER	Burdorf, Alex Erasmus MC, Department of Public Health
REVIEW RETURNED	24-Feb-2021

GENERAL COMMENTS	 1. The abstract lacks information on duration of trajectories (5 yr). 2. Awkward definition of injuries: road traffic injuries during commuting from home to work 3. The photovoice study is aimed to identify issues of importance..but on what? on timely RTW? 4. I still think that the control group is a bit confusing, since we know that young age is a risk factor for injuries and for sickness absence, hence, if the control group selects an other age group then two things are intermingled: age and injury. Thus, in my opinion there are two options: change the control group, or leave out the control group completely and focus on trajectories within those with an injury (but then select those variables that will influence the trajectories) 5. Analysis: some sentences lack clarity, eg dependent variable is mean number of sickleave days ? mean per year? it is truly a mean or just number of sickness days in a given period? 6. Analysis: I am still not sure what goal is of logistic regression and its outcome measure (RTW?). Given the rationale in the introduction, this comes a bit as a surprise. 5.
---

REVIEWER	Flachs, Esben Bispebjerg Hospital, Occupational and Environmental Medicine
REVIEW RETURNED	08-Mar-2021

GENERAL COMMENTS	Thank you for a nice revised version.
---------------------------------------

VERSION 2 – AUTHOR RESPONSE

Comments to the Author	Answer from authors
Reviewer 2 Prof. Alex Burdorf, Erasmus MC	
1. The abstract lacks information on duration of trajectories (5 yr).	The abstract has been revised accordingly. Please see page 2.

2. Awkward definition of injuries: road traffic injuries during commuting from home to work	The definition has been clarified (p. 7).
3. The photovoice study is aimed to identify issues of importance..but on what? on timely RTW?	The first sentence is a general description of the photovoice method. The sentence starting on the fifth row in the photovoice section describes which issues photovoice will address (see p. 6).
4. I still think that the control group is a bit confusing, since we know that young age is a risk factor for injuries and for sickness absence, hence, if the control group selects an other age group then two things are intermingled: age and injury. Thus, in my opinion there are two options: change the control group, or leave out the control group completely and focus on trajectories within those with an injury (but then select those variables that wil influence the trajectories)	Thank you for this suggestion. However, currently there is a lack of knowledge on the trajectories of young adults as mentioned in the background. One of the aims of the study is to assess how young adults' sick leave patterns differ from other age groups following a work injury. All participants included in the study have suffered a work injury.
5. Analysis: some sentences lack clarity, eg dependent variable is mean number of sickleave days ? mean per year? it is truly a mean or just number of sickness days in a given period?	A clarification has been added (p. 9)
6. Analysis: I am still not sure what goal is of logistic regression and its outcome measure (RTW?). Given the rationale in the introduction, this comes a bit as a surprise.	A clarification has been added (p. 9)

Comments to the Author Dr. Esben Flachs, Bispebjerg Hospital	Answer from authors
Thank you for a nice revised version.	Thank you for the input.

VERSION 3 – REVIEW

REVIEWER	Burdorf, Alex Erasmus MC, Department of Public Health
REVIEW RETURNED	31-Mar-2021
GENERAL COMMENTS	It seems to continue to disagree on appropriate control group, but if the aim is to compare trajectories across age groups, then I have no further comments.